# IRF8: Mechanism of Action and Health Implications

**DOI:** 10.3390/cells11172630

**Published:** 2022-08-24

**Authors:** Hannah R. Moorman, Yazmin Reategui, Dakota B. Poschel, Kebin Liu

**Affiliations:** 1Department of Biochemistry and Molecular Biology, Medical College of Georgia, Augusta, GA 30912, USA; 2Georgia Cancer Center, Augusta, GA 30912, USA; 3Charlie Norwood VA Medical Center, Augusta, GA 30904, USA

**Keywords:** IRF8, myeloid cells, OPN, MDSCs, immunotherapy

## Abstract

Interferon regulatory factor 8 (IRF8) is a transcription factor of the IRF protein family. IRF8 was originally identified as an essentialfactor for myeloid cell lineage commitment and differentiation. Deletion of *Irf8* leads to massive accumulation of CD11b^+^Gr1^+^ immature myeloid cells (IMCs), particularly the CD11b^+^Ly6C^hi/+^Ly6G^−^ polymorphonuclear myeloid-derived suppressor cell-like cells (PMN-MDSCs). Under pathological conditions such as cancer, *Irf8* is silenced by its promoter DNA hypermethylation, resulting in accumulation of PMN-MDSCs and CD11b^+^ Ly6G^+^Ly6C^lo^ monocytic MDSCs (M-MDSCs) in mice. IRF8 is often silenced in MDSCs in human cancer patients. MDSCs are heterogeneous populations of immune suppressive cells that suppress T and NK cell activity to promote tumor immune evasion and produce growth factors to exert direct tumor-promoting activity. Emerging experimental data reveals that IRF8 is also expressed in non-hematopoietic cells. Epithelial cell-expressed IRF8 regulates apoptosis and represses Osteopontin (OPN). Human tumor cells may use the IRF8 promoter DNA methylation as a mechanism to repress IRF8 expression to advance cancer through acquiring apoptosis resistance and OPN up-regulation. Elevated OPN engages CD44 to suppress T cell activation and promote tumor cell stemness to advance cancer. IRF8 thus is a transcription factor that regulates both the immune and non-immune components in human health and diseases.

## 1. Introduction

Interferon regulatory factor 8 (IRF8), originally termed interferon consensus sequence binding protein (ICSBP), is a member of the IRF transcription factor [1]. IRFs were first identified as a regulator of the type I interferon (IFN-I) response for the activation of IFN-stimulated genes that are essential for immune response to viruses and other pathogens [2,3,4,5], and are now known to be important in turning pathogen associated molecular patterns into chromatin changes and eventually into immune cell activation [6]. IRF8 was first cloned as an IFNγ-inducible nuclear protein-encoding gene that binds to specific IFN-responsive DNA motif in the major histocompatibility complex class I (MHC I) genes [7]. IRF8 has since been determined to be constitutively expressed and IFNγ inducible and plays key roles in the IFN response pathways in immune cell differentiation and function, as well as in non-hematopoietic cell turnover and pathogenesis [8,9,10,11,12,13,14,15,16,17,18,19,20,21].

## 2. IRF8 Function and Diseases

A 915C> T mutation (R294C) resembles IRF8 KO mice in accumulating immature myeloid cells (IMCs) and causes susceptibility to infection in mice [22,23]. K108E and T80A mutations results in IRF8 loss of function that leads to impaired dendritic cell monocyte development and function in humans or mice [9,24]. A 331C>T [R111* (stop codon)] mutation may cause IRF8 loss of function, resulting in neutrophilia, monocytopenia and decreased CD3^+^ T cell and CD8^+^ T cell counts in humans [25]. IRF-8 polymorphisms have been implicated in the development of autoimmune thyroiditis, Behcet’s disease and, increased susceptibility to tuberculosis (TB). The IRF-8 polymorphism, rs17445836, was associated with both development of autoimmune thyroiditis and TB [26,27,28]. Moreover, IRF8 also functions as an apoptosis regulator in hematopoietic tumor cells [29,30,31,32]. IRF8’s intrinsic function in hematopoietic cells thus plays essential roles in myelopoiesis to regulate immune cell activation and immune suppressors under physiological conditions [20,33,34] and in suppression of tumorigenesis [29,30,31,32]. Indeed, IRF8 has been shown to function as a suppressor of acute myeloid leukemia (AML) [35]. However, IRF8 has recently been reported to function as an acute myeloid leukemia (AML)-specific susceptibility factor and promote AML cell proliferation. High IRF8 expression is associated with poorer prognoses in certain AML patients [36]. On the other hand, IRF8 is also expressed in non-hematopoietic cells [8,10,37,38,39,40,41,42,43,44,45]. IRF8 functions as an apoptosis regulator in several types of human tumor cells [38,41,46,47] and mice with IRF8 deletion only in the colon epithelial cells develop significantly more colon tumors than the littermate control mice [41]. Furthermore, IRF8 expression level is positively linked to cancer patient prognosis and response to immunotherapy [18,19]. IRF8 therefore also functions as a tumor suppressor in non-hematopoietic cells. In this review, we discuss the function of IRF8 in hematopoietic and non-hematopoietic cell populations and the role of IRF8 in antitumor immune responses. 

## 3. IRF8 Function as a Transcription Factor That Depends on IAD-Interacting Transcription Factors to Exert Its Activity

Analysis and comparison of the IRF8 protein sequences in the National Center for Biotechnology Information with the Blast protein program [48] revealed that IRF8 proteins are highly conserved in mammals, with an 89.44–89.67% similarity in amino acid sequences between human and mouse IRF8 proteins. Human IRF8 gene is located at chromosome 16 with 23,228 bp that encodes 9 exons and 8 introns (Figure 1). The human IRF8 protein is 426 amino acids long with a DNA-binding domain (DBD) and an IRF association domain (IAD) [23]. The DBD binds to unique consensus sequence motifs at promoters of genes, such as IFN-responsive genes, to regulate transcription. The IAD associates with other transcription factors to direct the IRF8 protein complex binding to specific DNA motif to regulate specific gene transcription (Figure 1). IRF8 has weak DNA-binding affinity and its transcription regulatory activity therefore depends on its IAD association with other transcriptional factors to form a transcription complex which renders IRF8 specific and high affinity to bind to the unique DNA motifs at gene promoters [23,34,49,50]. IRF8 functions as either a transcriptional activator or repressor and it is the IAD-associated transcription factor(s) that dictates IRF8 bindingand transcriptional activity [20,23,33,51,52,53,54,55].

IRF8 transcription factor protein complexes selectively binds to certain types of DNA consensus sequence motifs to activate or repress gene transcription [33,56,57,58,59]. Heterodimers of IRF8 in association with IRFs (e.g., IRF1 or IRF2) bind to the IFN-stimulated response element (ISRE: [(A/G)NGAAANNGAAACT]. IRF8-IRF1/IRF2 heterodimers generally bind to promoter region ISRE to repress the transcription of the downstream genes (Figure 1) [1]. The IRF8-Ets (Erythroblast Transformation Specific/E-twenty-six) and IRF8-PU1 heterodimers bind to the Ets-RF8 composite element (EICE: GGAANNGAAA) or the IRF8-Ets composite sequence (IECS: GAAANN(N)GGAA) to activate target gene transcription [52,53,56,57,58,60,61,62,63,64,65,66]. In addition, IRF8 has been shown to form a complex with the JUNB/AP-1-BATF3and/or IRF4 and binds to the AP-1-IRFcomposite elements 1 and 2 [AICE1:TTTCNNNNTGA(G/C)T(C/A)A; AICE2: GAAATGA(G/C)T(C/A)A] to activate transcription of genes [11,67,68,69]. For example, IRF8 forms a heterodimer with AP-1 factor BATF and binds to the AICE site to promote gene activation in T helper 17 (Th17) cells, B and dendritic cells [67,69,70]. It has been shown that IRF8 also forms complexes with IRF4, PU.1 and BATF at the ISRE-like sites to activate *Il9* and *Il21* transcription in Th9 cells [66]. IRF8 can also form a heterodimer with IRF1 or IRF2 [71] that binds to ISRE element to repress transcription of genes induced by IFN and retinoic acid [72,73] or to activate target gene expression [50,70,71]. Although IRF8 binds to IECS to activate gene expression [1,54], IRF8 binding to IECS element at the *Asah1* promoter represses acid ceramidase expression [31]. Furthermore, IRF6 interacts with ETV6 to repress *Il4* transcription in Th9 cells [66].

## 4. IRF8 Expression Profiles

The expression of IRF8 was originally detected in hematopoietic cells and its expression has long been thought to be restricted to hematopoietic cells [56]. Although IRF8 is abundantly expressed in hematopoietic cells, particularly in B cells and DCs, it has since been determined that IRF8 is also expressed in epithelial cells in the intestine, colon, ocular lens, skin, lung, liver, and heart [8,10,37,38,39,40,41,42,43,44,45]. Analysis of IRF8 expression from single cell RNA sequencing datasets revealed that IRF8 is indeed expressed in several types of non-hematopoietic cells, including theca cells, melanocytes, enterocytes and intestinal cells in humans (Figure 2).

## 5. IRF8 Function in Hematopoietic Cells

In hematopoietic cells, the highest IRF8 expression is in the progenitors and mature cells of the B cell, cDC1, and pDC lineages [1,20,74,75]. Consistent with the high level of IRF8 expression in these cell lineages, IRF8 is essential for the development of monocytes and DCs (Figure 3). IRF8 may also regulate myeloid cell lineage differentiation indirectly through regulating GM-CSF expression in T cells [76]. Loss of IRF8 expression or function often leads to impaired immune response to infection and hematopoietic malignancies [9,22,29,30,31,77,78]. IRF8 deficiency results in a severe immunodeficiency humans as characterized by susceptibility to infections due to loss of DC subsets, CD14^+^ and CD16^+^ monocytes and a decreased level of NK cells with reduced activity [70]. A genome-wide ChIP-Seq analysis identified 319 IRF8-binding sites at the gene promoter regions with an EICE-like motif in macrophages stimulated with IFNγ/CpG. Analysis of these IRF8 target genes in *Mycobacterium tuberculosis*-infected lungs revealed specific enrichment of pathways/genes in antigen processing and presenting, indicating a key role of IRF8 function in myeloid cells in regulating early response to infection [79]. De novo DNA motif analysis using IRF8 ChIP-seq identified 89 genes with an AICE-dependent gene program that requires high levels of IRF8 in cDC1 [75]. These findings indicate that IRF8 directly regulates a large set of genes in myeloid cells.

IRF8 also plays a key role in regulating Th1 polarization of early immune response following infection. IRF8-deficient mice exhibit diminished production of IL-12p40, lack of Th1 polarization [1,24,70]. Mice with deletion of the *Irf8* gene exhibit a significantly reduced number of pre-pro-B cells. Hematopoietic stem cells of IRF8 KO mice are deficient in differentiation to B220^+^ B-lineage cells and this defect can be corrected by adoptive transfer of bone marrow cells of WT bone marrow cells [80,81,82]. IRF8 therefore not only plays an essential role in the linage differentiation of innate immune cells but also plays a key role in the development of the adaptive immune cells.

### 5.1. Dendritic Cells

Dendritic cells (DCs) are found in most tissues and classically function to activate antigen-specific naïve T cells. Discoveries of different transcription factors and surface markers allow for distinction between DCs and other immune cells as well as distinction between different DC subsets. DCs are divided into either conventional (cDC1 and cDC2) or plasmacytoid (pDC). Conventional DCs prime naïve T cells according to their surface markers (cDC1 cells present to CD8^+^ T cells and cDC2 cells present to CD4^+^ T cells) while pDCs produce large quantities of Type I interferons [83,84] and their antigen-presenting ability is likely less efficient than that of conventional DCs [85]. Much of the research regarding IRF8 in the context of DCs is directed at understanding its role in differentiation [86,87,88].

In the early stages of DC differentiation, IRF8 is required for the transition from macrophage and DC precursor (MDP) to common DC precursor (CDP) [89,90]. Not all the literature is in agreement about the point at which IRF8 becomes necessary for DC differentiation. Another study found that at an earlier lymphoid primed multipotent progenitor (LMPP) stage, IRF8 expression biased differentiation towards DCs by making DC-lineage genes more accessible [91]. When IRF8 was absent from MDPs, differentiation was skewed in favor of a macrophage phenotype rather than a DC phenotype [90]. Additionally, IRF8 was required for differentiation into cDC1 cells but not required for the differentiation of cDC2 cells from the CDP stage [89]. This indicates that IRF8 is necessary for commitment to the DC lineage but is more involved in cDC1 speciation than cDC2 speciation. As for pDCs, IRF8 was not necessary for their differentiation from CDPs. However, IRF8 KO mice produced less Type I IFN in response to stimuli indicating that IRF8 might play a larger role in the function of pDCs than in their development [89].

Recently there has been increasing evidence for an association between IRF8 and dendritic cell deficiencies leading to diseases such as chronic myelogenous leukemia [92], atherosclerosis [93], and cancers [42,94]. For example, IRF8-deficient mice display a CML- like phenotype dominated by expansion of granulocyte precursors and lack both CD8α^+^ dendritic cells (DCs) and plasmacytoid DCs (pDCs) [70]. An autosomal recessive IRF8 deficiency, K108E, was associated with a complete lack of dendritic cells and resulted in a severe immunodeficiency [9]. Upon analysis of this patient’s blood, the IRF8-deficient dendritic cells exhibited defective antigen presentation and anergic T cells. IFR8 is responsible for survival and function of terminally differentiated cDC1s. It has been shown that increased IRF8 expression in CD11b^−^CD103^−^ DN DC progenitor cells promotes differentiation into cDC1s to enhance CAR-T cell efficacy [95]. Because IRF8 is necessary for DC differentiation and adequate functioning, the loss of IRF8 may provide an immunodeficient environment contributing to disease pathogenesis and progression.

### 5.2. Macrophages

IRF8 is involved in the differentiation of macrophages. Myeloid progenitor cells established from IRF8-deficient mice were able to differentiate into functional macrophages and express macrophage-specific genes upon retroviral transduction of IRF8 [96]. In contrast, transduction of IRF8 resulted in a decrease in granulocytic differentiation and repression of granulocytic differentiation genes. Together, this suggests that IRF8 promotes macrophage differentiation over granulocytic differentiation from myeloid progenitor cells. IRF8’s role in macrophage differentiation was also studied in a zebrafish embryogenesis model [97]. IRF8 knockout models showed a depletion of macrophages, an increase in neutrophils (a subset of granulocytes), but no difference in myeloid progenitors. Upon overexpression of IRF8, macrophage differentiation was promoted while neutrophil differentiation was suppressed. This model further supports the ability of IRF8 to promote macrophage differentiation over granulocyte differentiation.

IRF8 also appears to be involved in the homeostasis and differentiation of tissue macrophages including osteoclasts [13,17,98,99]. In IRF8-deficient mice, macrophage morphology and cell numbers were assessed. IRF8-deficient macrophages from the brain, liver, and kidney exhibited changes in their population sizes; in brain and liver macrophages, cell morphology was changed as well. For macrophages to maintain their functionality, different genes related to immune functioning must be expressed. In IRF8-deficient macrophages, a downregulation of immune-related genes like *toll-like receptor 12* and *major histocompatibility complex class II* was observed. This study indicates that IRF8 plays a role in maintaining the morphology, population size, and gene profile of tissue macrophages [98].

IRF8 regulates macrophage functioning as part of the innate immune system. It promotes transcription of pro-inflammatory cytokines such as IL-12, affects macrophage responses to LPS stimulation, and determines macrophage response to intracellular and engulfed pathogens. In IRF8-deficient mice, increased susceptibility to intracellular infection was due to impaired synthesis of IL-12 [100]. Co-transfection with IRF8 and IRF1 was found to act synergistically to activate the IL-12 promoter as determined by a luciferase reporter gene activity [101]. Additionally, IRF8 and IRF1 were found to act through binding to an IRSE-like sequence within the IL-12 promoter. Taken together, IRF8 (acting synergistically with IRF1) activates the IL-12 promoter through binding with an IRSE-like sequence. Enhanced IL-12 allows for more effective resistance to intracellular infection.

In addition to IL-12 production, other IRF8-mediated macrophage responses to bacterial infection include autophagy and inflammasome activation. IRF8 was found to activate genes involved in autophagosome formation and lysosome fusion [102]. As such, in IRF8-deficient macrophages, activation of autophagy genes and pathogen degradation was defective and growth of the facultative intracellular pathogen, *Listeria monocytogenes*, was unhindered. Similarly, IRF8 induced *Nramp1* (cation transporter involved in phagosome functioning) expression through activation of its promoter [103]. Decreased clearance of intraphagosomal pathogens was observed in macrophages deficient in IRF8 or Nramp1, suggesting that IRF8 induces Nramp1 thereby facilitating phagosome degradation of pathogens. Within macrophages, IRF8 was found to promote NLRC4 inflammasome activation by enhancing the transcription of *Naips* [104]. Additionally, IRF8 was found to protect against *S. typhimurium* and *B. thailandensis* bacterial infection in vivo likely due to inflammasome-dependent cytokines and pyroptosis. These studies highlight the role of IRF8 in the macrophage response to pathogens.

### 5.3. Monocytes

Differentiation in monocytes is regulated by IRF8. In one study, granulocyte progenitors and monocyte progenitors were able to be separated from their common granulocyte/monocyte progenitor through cell sorting [105]. Higher IRF8 expression was observed in the lineage-committed cell lines rather than in their common progenitor. IRF8 KO in vitro demonstrated that differentiation of granulocyte progenitors and monocyte progenitors from their common granulocyte/monocyte progenitor was not affected by a lack of IRF8, indicating that IRF8 does not work at the level of granulocyte/monocyte progenitors, but is rather involved in lineage-specific differentiation. Another study found that in IRF8 KO mice, mononuclear phagocyte progenitor cells were unable to give rise to monocytes and instead gave rise to neutrophils [34]. Mechanistically, IRF8 was found to interact with transcription factor C/EBPα and block its ability to stimulate transcription of the genes necessary for neutrophil differentiation, thereby suppressing the neutrophil phenotype. Indeed, a previous study indicated that IRF8 KO mice were found to lack inflammatory Ly6C+ monocytes [57]. Similarly, another group found that IRF8-deficient mice infected with West Nile virus exhibited reduced numbers of peripheral Ly6C+ monocytes with diminished migratory capabilities [106]. Taking the results of these studies together, IRF8 appears to promote monocyte differentiation into specific subsets and suppress neutrophil differentiation.

IRF8 expression and its subsequent role in monocyte differentiation appear to be regulated by the signal transducer and activator of transcription (STAT) protein family, specifically STAT1 [107] and STAT5 [108]. When all-trans retinoic acid (ATRA), which promotes myelomonocytic differentiation, was given to human monoblastic cells, up-regulation of IRF8 (and other differentiation transcription factors) was impaired in cells expressing mutant STAT1 and monocytic inhibition was impaired [107]. This indicates that STAT1 is involved in the upregulation of IRF8 in monocytes thereby contributing to differentiation. In a study on the effects of mTOR on monocyte differentiation, blocking STAT5 activity was found to increase IRF8 expression and increase the differentiation in mTOR-deficient granulocyte/monocyte precursors [108]. This study suggests that STAT5 negatively regulates IRF8 expression. While it is known that ATRA induces STAT expression to promote differentiation in myeloid leukemia [109], one such mechanism may be through the regulation of IRF8 expression.

### 5.4. IMCs/Myeloid-Derived Suppressor Cells (MDSCs)

A major phenotype of IRF8 knock out (IRF8 KO) mice or mice with IRF8 deficiency is accumulation of CD11b^+^Gr1^+^ IMCs [23,110] that phenotypically and functionally resembles tumor-induced CD11b^+^Gr1^+^ myeloid-derived suppressor cells (MDSCs) [32,111]. Among the MDSC subtypes, CD11b^+^Ly6C^hi/+^Ly6G^–^ polymorphonuclear MDSCs (PMN-MDSCs) is dramatically increased in IRF8 KO mice. Although the percentages of monocytic MDSCs (M-MDSCs) were similar to the WT mice, there is a significant increases in the total numbers of both PMN-MDSCs and M-MDSCs in IRF8 KO mice [111]. In tumor-bearing mice, IRF8 expression level is silenced in both PMN-MDSCs and M-MDSCs [111]. IRF8 is also down-regulated in MDSCs of human cancer patients [32,112]. IRF8 expression can be repressed by its promoter DNA methylation [41]. IRF8 expression is also regulated by micro-RNA, phosphorylation, and ubiquitination [113,114,115]. Under non-pathological conditions, immature myeloid cells differentiate into granulocytes, macrophages, and DCs. However, pathological conditions such as cancer can block their differentiation and increase the expression of immune-suppressive factors. The resulting cells are considered MDSCs [116,117,118,119].

Between the two subsets of MDSCs, PMN-MDSCs exhibit more extensive expansion in cancer when compared to M-MDSCs [120,121], and a decrease in IRF8 has been associated with an increase in MDSC frequency in cancer [111]. Recent research supports IRF8’s role in inhibiting the formation of MDSCs. Tumor growth in mice was associated with the selective expansion of low-IRF8 granulocyte progenitors (GPs). Importantly, tumor-derived granulocyte progenitors exhibited an increased ability to form PMN-MDSCs [122]. When gene expression profiles were compared between tumor-derived GPs and IRF8 KO GPs, shared expression profiles indicated that loss of IRF8 may underlie the formation of GPs and subsequent PMN-MDSC formation in tumors. Further supporting MDSC suppression by IRF8, IRF8-deficient mice generated MDSC-like populations highly homologous to tumor-induced MDSCs while IRF8 overexpression attenuated MDSC accumulation [111].

While IRF8 inhibits the formation of MDSCs, MDSCs (and other factors) also negatively regulate the expression of IRF8. Chronic inflammation, such as in the setting of ulcerative colitis, increases the population of MDSCs. MDSCs express large amounts of IL-10 which activates transcription factor STAT3. Through the activation of DNA methyltransferases, IRF8 is functionally silenced [41]. Taken together, this suggests that MDSC elevation can silence IRF8 expression consequently increasing the expansion of MDSCs. Decreased IRF8 in the context of MDSCs promotes their survival through evasion of cytotoxic T lymphocyte (CTL) elimination. MDSCs can alter the expression of apoptotic proteins Bax and Bcl-xl and deregulate the Fas-mediated apoptosis program utilized by CTLs by suppressing IRF8 [32]. As a result, MDSCs can evade immune clearance [117,123].

### 5.5. B Cells

Originally IRF8 was found to be expressed in mature mouse B cells and B cell precursors (pro and pre-B cells), while not expressed in tumors composed of mature plasma cells [124]. Further research has supported the differential expression of IRF8 between B cells and terminally differentiated B cells, known as plasma cells, and shown that IRF8 promotes germinal center (GC) formation while altering B cell population [82,125,126,127,128,129]. IRF8 promotes the GC B cell response while inhibiting plasma cell differentiation through the induction of genes that maintain the B cell program and repression of genes that maintain the plasma cell program [130]. Actions that promote the GC B cell response such as dampening of BCR signaling, facilitation of B cell-T cell interactions, and promotion of affinity maturation are regulated by IRF8. In addition to promoting the GC B cell response, IRF8 also antagonizes the IRF4-driven differentiation of plasma cells [81]. Double deletion of both IRF8 and PU.1 led to enhancement in the rate of class switch recombination and plasma cell differentiation further supporting the role of IRF8 in the negative regulation of plasma cell differentiation. It was recently found that mice with deletion of *irf8* in B cells have abundance of regulatory B cell (i35-Bregs) [131].

IRF8 has been implicated in diffuse large B cell lymphoma (DLBCL) and pre-B cell acute lymphoblastic leukemia (Pre-B ALL) [132,133,134]. In DLBCL, IRF8 was discovered as an immunoglobulin heavy chain (IGH) fusion partner. Expectedly, the lymphomas with this rearrangement exhibited markedly higher IRF8 expression. IRF8-regulated actions such as induction of genes that promote B cell formation, suppression of plasma cell formation, and promotion of apoptosis resistance likely contribute to the formation of DLBCL [132]. This is in agreement with earlier research suggesting that IRF8 decreases the expression of p53 and p21 facilitating an increase in apoptosis resistance [135]. A proportion of mice double deficient in PU.1 and IRF8 developed Pre-B ALL. It was shown that IRF8, PU.1, and IRF4 bound to the promoter/enhancer regions of tumor suppressor genes, such as Ikaros and Spi-B, and in IRF-deficient samples, the expression of these genes was significantly reduced [133]. While it appears that IRF8 protects against the formation of Pre-B ALL, drastically increased IRF8 (such as in the case of IGH-IRF8 fusion) may promote the formation of DLBCL.

### 5.6. T Cells

Much of the research on IRF8 function in T cells focuses on its role in T cell differentiation. T Helper 17 (Th17) cells are a subset of CD4+ Helper T cells named for their secretion of cytokine IL-17. IRF8 has been reported to suppress the expansion of Th17 cells [8,136,137] although not all literature is in agreement with this [138]. IRF8 appears to inhibit the expression of RORγt, a crucial transcription factor for Th17 cell differentiation [136,139,140]. Additionally, overexpression of IRF8 suppressed activity at a RORγt-dependent enhancer required for IL-17 expression [137].

IRF8 involvement in the differentiation of other T cell lineages has also been investigated. IRF8 may facilitate the differentiation of T Helper 1 (Th1) cells over T Helper 2 (Th2) cells. IRF8 and IRF1 expression were higher in Th1 polarized cells versus Th2 polarized cells while IRF4 expression was higher in Th2 polarized cells [141]. This implicates the differential expression of IRFs in Helper T cell differentiation with IRF8 being most likely involved in Th1 differentiation. This is congruent with earlier research finding that IRF8 was required for adequate Th1-induced macrophage activation [142]. IL-9 producing Helper T (Th9) cells exert an anti-tumor effect through the secretion of IL-9 and IL-21 [143,144,145]. IRF8 was shown to be required for the differentiation of Th9 cells through binding to Th9 related genes and suppressing the secretion of IL-4 [66].

IRF8 has a relatively unclear role in CD8^+^ T cells. IRF8 may restrain the expansion of CD8^+^ T cells. In a mouse model of ocular herpes simplex 1 virus (HSV-1) infection, IRF8 KO mice exhibited marked expansion of HSV-1 specific CD8^+^ T cells with increased infiltration, inflammation, and viral clearance [40]. This suggests that IRF8 is involved in the suppression of CD8^+^ T cells and dampening of the associated inflammation. However, a previous study found that depletion of IRF8 decreases differentiation of CD8^+^ T cells into effector T cells in a mouse model of Graft Versus Host Disease [146].

IRF8 suppresses inflammation and is involved in tumor formation. A decrease in IRF8 increased inflammation in a mouse model of colitis possibly due to the associated increase in Th17 cells [137]. A loss of IRF8 also resulted in increased inflammation in autoimmune uveitis (associated with a significant increase in Th17 cells) and Ocular HSV-1 infection (possibly due to a decreased limitation on CD8^+^ expansion) [8,40]. Decreased IRF8 expression in CD8^+^ T cells has been associated with an increase in melanoma growth while an increase in IRF8 has been reported in DLBCL tumor tissues [66,136]. In DLBCL, IRF8-induced reduction of Th17 expansion may contribute to tumor formation.

### 5.7. Natural Killer Cells

IRF8 is involved in the development and function of Natural Killer (NK) cells [147]. Biallelic mutations in IRF8 resulted in impaired NK cell function, numbers, and maturation [148,149]. It has been shown that T cells and natural killer cells are primary sources of IFN-*γ* production, but it is unclear whether these two cell types are the main source of IFN-*γ* in response to CpG oligodeoxynucleotides. It was revealed that IRF8 directly regulates the expression of TLR9 and its deficiency leads to no increased IFNγ production expected with CpG stimulation [147]. Physiologically, IRF8 is highly expressed in NK cells. It was shown that IRF8-deficient mice exhibited a decreased proportion and total count of mature NK cells in the bone marrow. This was due to elevated neutrophil abundance in *Irf8^−/−^* bone marrow cavity [147]. In addition, because NK cells are involved in the early immune response against viral pathogens, the resulting NK cell deficiencies resulted in severe viral disease. Similarly, NK IRF8 -/- mice infected with MCMV exhibited poorly controlled viral replication and decreased survival [150]. IRF8 -/- NK cells exhibited decreased virus-driven expansion and impaired proliferation which may account for the poorly controlled viral replication seen in the mice models of IRF8 deficiency [150].

## 6. IRF8 Expression and Function in Non-Hematopoietic Cancer Cells

While much research has been dedicated to understanding the role of IRF8 in hematopoietic cells, less research has investigated the role of IRF8 in non-hematopoietic cells. In certain types of cancers, IRF8 functions as a tumor suppressor gene [18,19,42,43,44], and its expression is regulated by DNA methylation or micro-RNA interference [113]. In breast and lung cancer, IRF8 is downregulated as a result of promoter hypermethylation [43,151,152], and a higher IRF8 expression level was associated with a better prognosis [19]. Similarly, in gastric, nasopharyngeal, esophageal, colorectal, lung, prostate, and renal cell carcinomas the expression of IRF8 is downregulated [43,44,153,154,155,156,157]. IFN-γ induced expression of IRF8 was increased when gastric cancer cells were treated with a demethylating agent [153]. In other cancer cell lines with a methylated IRF8 promoter, IFN-γ treatment showed decreased ability to induce IRF8 expression [154]. Taken together, these studies provide evidence for promoter methylation as a mechanism for IRF8 silencing in certain types of human cancer cells. Analysis of human lung tumor specimens at the single cell level validate the early finding that IRF8 is silenced in the tumor cells. Except for certain tumor-infiltrating immune cells, including certain B cells, pDC and myeloid cells (Figure 4A), IRF8 is down-regulated in most cell types and is undetectable in human lung cancer cells (Figure 4B) [158].

Within non-hematopoietic cancer cells, IRF8 plays a protective role against the formation of a metastatic phenotype. In human colon cancer cells, IRF8 protein levels are inversely correlated with the metastatic phenotype [46,47]. IRF8 expression sensitized the metastatic tumor cells to Fas-mediated apoptosis, indicating that loss of IRF8 in cancers could promote earlier metastasis [46]. It has been suggested that colon cancer cells silence IRF8 expression through inhibition of pSTAT1 function at the IRF8 promoter [47]. IRF8 may also function in non-hematopoietic cancers to negatively regulate the expression of MMP3, a matrix metalloproteinase associated with a poor prognosis and metastasis in various solid organ cancers [159,160,161]. In a chemically-induced sarcoma model, an inverse relationship between IRF8 and MMP3 expression was demonstrated; in a mouse model of mammary cancer, loss of MMP3 reduced spontaneous lung metastasis [162]. This suggests that IRF8 may negatively regulate MMP3 expression, protecting against MMP3-induced metastasis in solid organ tumors. While most research supports an anti-metastasis role for IRF8 in non-hematopoietic cancers, one study found that IRF8 promoted EMT-like phenomena, cell motility, and invasion in a human osteosarcoma cell line providing evidence for a possible role in the acquisition of a metastatic-like phenotype [163].

IRF8 appears to impede the progression and formation of cancer by increasing the expression of tumor suppressor genes, such as Caspase 1, p21, p27, and PTEN [44,157,164]. Expression of tumor suppressor genes inhibited lens cell carcinoma growth upon transfection with IRF8 [164]. Similarly, IRF8-induced expression of p27 induced lung cancer cell senescence [44]. Mechanistically, IRF8 was shown to inhibit Akt activity thereby inducing the accumulation of p27 and promoting senescence. While IRF8 increases tumor suppressor genes, one study also showed an IRF8-induced decrease in oncogene (Yap1 and Survivin) expression in renal cell carcinoma cell lines [157]. In this cell line, ectopic expression of IRF8 induced cell cycle G2/M arrest and apoptosis further supporting IRF8’s role in impeding cancer progression.

IRF8 functions as a promoter of Fas-mediated apoptosis. This form of extrinsic apoptosis is key to immune clearance of tumor cells and is targeted in cancer therapy [165,166]. Modification of Fas expression can render immune checkpoint inhibitor (ICI) therapies unsuccessful. Colon cancer cells with low Fas expression exhibited decreased sensitivity to FasL-induced apoptosis (a mechanism utilized by ICIs) [167] and lower Fas expression was correlated with decreased survival in colon cancer patients [168]. One mechanism by which cancer cells may alter the expression of Fas, and thereby reduce sensitivity to apoptosis, is through downregulation of IRF8.

As previously mentioned, IRF8 is downregulated in many cancers. Studies revolving around a loss of IRF8 have elucidated ways in which IRF8 helps facilitate Fas-mediated apoptosis and resulting in tumor immune sensitivity. In soft tissue sarcoma cells, IRF8 was found to be a repressor of FLICE-like protein (FLIP) [169]. Silencing of FLIP and ectopic IRF8 expression each increased susceptibility to FasL-induced apoptosis. This suggests that by repressing FLIP, IRF8 promotes Fas-mediated apoptosis susceptibility. Mechanistically, IRF8 disruption diminishes JAK1 expression and inhibits STAT1 phosphorylation to block IFN-γ induced Fas upregulation in sarcoma cells [170]. IRF8 has also been found to regulate pro-apoptotic and anti-apoptotic factors contributing to Fas-mediated apoptosis. In tumor-induced MDSCs, levels of IRF8 are decreased. IRF8-deficient MDSCs showed an increase in anti-apoptotic Bcl-xL and a decrease in pro-apoptotic Bax, providing a mechanism by which IRF8 downregulation can impede the Fas-mediated apoptosis pathway to evade elimination by T lymphocytes [32]. Another mechanism by which IRF8 can promote Fas-mediated apoptosis is through repression of acid ceramidase (A-CDase). A-CDase upregulation has been implicated in certain cancers such as prostate and breast [171,172,173]. IRF8 induced repression of A-CDase resulted in C16 ceramide accumulation and increased apoptosis sensitivity [31]. This suggests that IRF8 downregulation likely contributes to decreased Fas-mediated apoptosis sensitivity through failure to repress A-CDase. In summary, IRF8 promotes Fas-mediated apoptosis through repression of FLIP and A-CDase and by regulating apoptotic factors such as Bcl-xL and Bax [31,32,169].

IRF8 functions as a link between cancer and the immune system. Poor tumoral homing of DCs and T cells was observed in melanoma-bearing IRF8 KO mice [42]. Additionally, the few DCs that were able to infiltrate the tumor were immature and unable to produce T cell-mediated immune responses. Mechanistically, the decreased tumoral immune infiltration observed in IRF8 KO mice is suggested to occur through the aberrant expression of multiple chemokine receptors and ligands [42]. When IRF8 was induced, immune infiltration was reestablished, and the chemokine expression pattern was reversed. Using a microfluidic device, in which melanoma cells and splenocytes were separated by a microchannel, IRF8 KO splenocytes exhibited a marked decrease in the ability to cross the channel to infiltrate the melanoma cells while melanoma cells exhibited marked propensity to invade the channels suggesting existence of an IRF8-regulated communicating soluble factor [174].

OPN has recently emerged as another immune checkpoint that may compensate PD-L1 function to promote colon tumor immune evasion [175]. IRF8 is expressed in human and mouse gastric and colon epithelial cells [38,41] and binds to the ISRE elements at the *Spp1* promoter to repress OPN expression [176]. In human colon and pancreatic cancers, IRF8 expression is down-regulated and OPN is up-regulated [175,176,177,178,179]. scRNA-Seq analysis indicates that OPN is abundantly expressed in tumor cells, MDSCs, and ILCs [175]. Furthermore, scRNA-Seq analysis revealed the tumor-promoting role of SPP1^+^ cells in cancer [180,181]. OPN protein level is highly elevated in the peripheral blood in human cancer patients and OPN expression is up-regulated in several different types of human cancers [182,183,184,185,186,187,188,189,190]. Silenced IRF8 expression in both tumor cells and tumor-induced myeloid cells by DNA methylation and H3K9me3 deposition at the *Irf8* promoter are likely major mechanisms underlying OPN elevation in cancer patients and tumor-bearing mice [41,46,47,154,191]. It was observed that IRF8 KO mice exhibited deficient generation of antigen-specific T cells and this deficiency was due to OPN engagement of CD44 to directly suppress T cell activation [176]. Accordingly, OPN blockade increased activation of tumor-specific T cells and suppressed tumor growth in vivo [175]. OPN is highly expressed in human pancreatic tumor cells and may bind to CD44 on tumor cells via autocrine and paracrine manners to promote pancreatic cancer stemness and progression [177,179]. Therefore, in addition to functioning as a repressor of *SPP1* in immune cells such as MDSCs and ILCs, IRF8 not only functions as an intrinsic tumor suppressor [41] but also suppresses tumorigenesis and progression through repressing OPN expression in tumor cells (Figure 5) [176].

## 7. Conclusions

IRF8 is a transcription factor that was identified as a member of the IFN transcription factor family more than three decades ago. IRF8 is essential for myelopoiesis and loss of IRF8 expression or function results in accumulation of CD11b^+^Gr1^+^ IMCs that phenotypically and functionally resemble tumor-induced MDSCs. IRF8 also plays a key role in immune cell homeostasis and turnover to prevent hematopoietic malignancies. IRF8 silencing by its promoter DNA methylation leads to accumulation of MDSCs in both human cancer patients and tumor-bearing mice. Although IRF8 is highly expressed in hematopoietic cells and it was once thought that its expression was restricted to immune cells, IRF8 expression and function has been extended to non-hematopoietic cells. IRF8 is now known to play a key role in epithelial cell turnover and functions as a tumor suppressor in solid tumors. Furthermore, IRF8 has been found to protect against immune evasion of tumor cells via the IRF8-OPN axis to control tumorigenesis and progression. IRF8 therefore is not only a key regulator of host response to infection, but is also a key regulator in host cancer immune surveillance and response to immunotherapy. In the therapeutic front, recent developments have been made in the relationship between IRF8 and immunotherapies. One such study focused on the role of IRF8 in response to anti-CD20 treatments such as rituximab and ofatumumab. IRF8 was necessary for CD20 transcription and anti-CD20 therapy efficacy [192]. Additionally, IRF8 is a transcriptional regulator of CD37 in DLBCL and regulates the efficacy of anti-CD37 pharmacotherapies [55]. Continued investigation on anti-PD-L1 therapy has shown that, in hepatocellular carcinoma, IRF8 enhances the response to treatment and suppressed progression [18]. IRF8 is therefore a therapeutic target in both hematopoietic and non-hematopoietic cancer.

## Figures and Tables

**Figure 1 cells-11-02630-f001:**
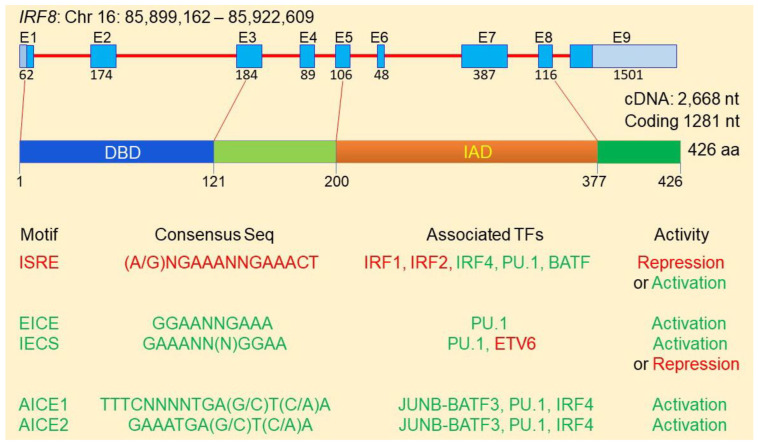
The IRF8 protein structure and transcription regulatory network.

**Figure 2 cells-11-02630-f002:**
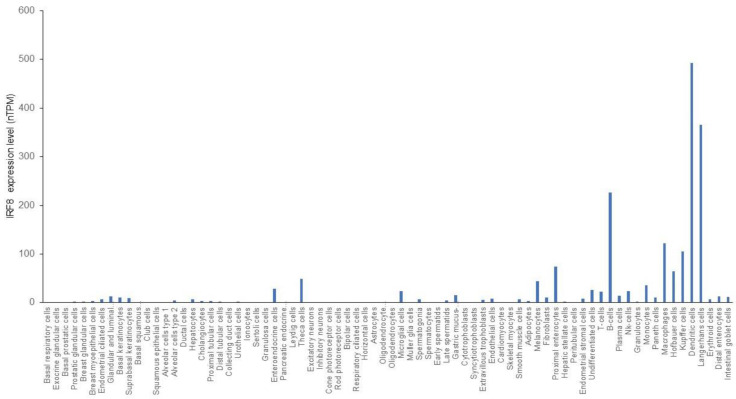
IRF8 expression profiles in human cells. Single cell RNA sequencing datasets were adapted from the human protein atlas.

**Figure 3 cells-11-02630-f003:**
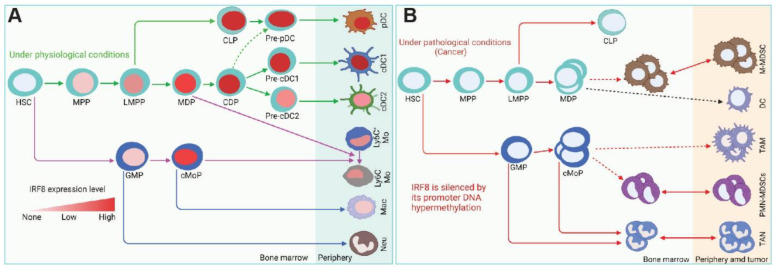
IRF8 functions under physiological and pathological conditions. (**A**). IRF8 regulation of myeloid cell lineage differentiation under physiological conditions. (**B**). Under pathological conditions such as cancer, IRF8 is silenced in myeloid cells, resulting in accumulation of immature myeloid cells and neutrophil. HSC: hematopoietic stem cell, MPP: multipotent progenitors, LMPP: lymphoid-primed multipotent progenitors, MDP: macrophage DC progenitors, CDP: common DC progenitors, CLP: common lymphoid progenitors, pDC: plasmacytoid DC, cDC1: conventional DC1, cDC2: conventional DC2, GMP: granulocyte-monocyte progenitor, cMoP: common monocyte progenitor, M-MDSC: monocytic-MDSC, PMN-MDSC: polymorphonuclear MDSC TAM: tumor-associated macrophage, TAN: tumor-associated neutrophils.

**Figure 4 cells-11-02630-f004:**
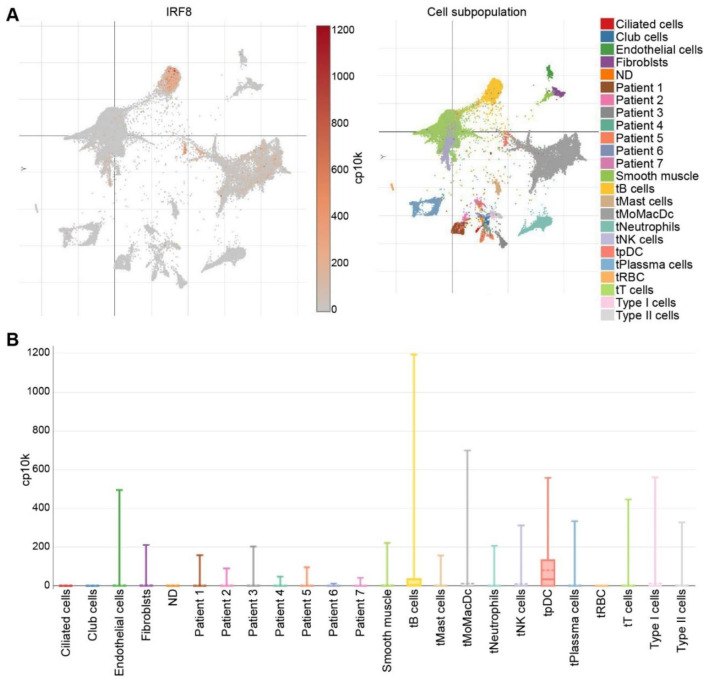
IRF8 expression profiles in human lung cancer in the single cell level. (**A**). scRNA-Seq datasets were extracted from the lung cancer scRNA-Seq datasets in the Broad Institute Single Cell Portal. Right panel shows UMAP plot of cell subpopulations from human lung tumor. Cells are color-coded. Left panel shows UMAP plot of IRF8 expression in the respective cells as shown in the right panel. (**B**). Box plot showing IRF8 expression level in the indicated cells and tumor (Patients 1–7). t: tumor.

**Figure 5 cells-11-02630-f005:**
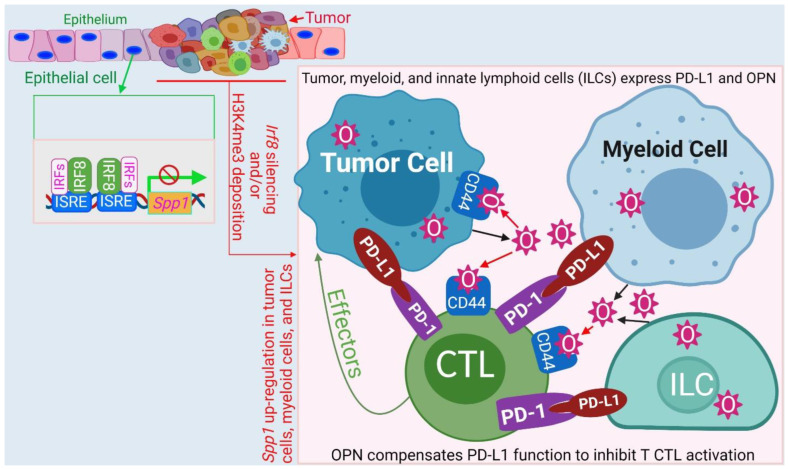
Model of the IRF8-OPN axis mechanism of action in immune suppression and tumor promotion. IRF8 functions as a repressor that binds to the two ISRE sites at the *Spp1* promoter to repress *Spp1* transcription in colon epithelial cells. During tumor development, *Irf8* is silenced by its promoter DNA methylation and H3K9me3 deposition, resulting in increased OPN expression in tumor cells, MDSCs, and ILCs. OPN binds to CD44 to suppress T cell activation and to CD44 on tumor cells to promote tumor cell stemness and progression. The IRF8-OPN axis thus control tumor growth and progression through its functions in both the immune and non-immune cell components.

## Data Availability

Not applicable.

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
