# Peer review of "IRF8: Mechanism of Action and Health Implications"

_cells, 2022, doi:10.3390/cells11172630_

Round 1

Reviewer 1 Report

In this review of Moorman et al. the authors describe the role of IRF8 in hematopoetic and non-hematopoietic cells. The provides a nice overview about the role of IRF8 in various context. However, the text often appears like an patchwork of not well connected statements and is not always easy to follow. Sometimes the review contradicts itself. For examples at the beginning it is written that IRF8 is not expressed in cancer cells, but later it is stated that IRF8 is often highly epressed in DLBLC. The review may therefore profit from some rewriting. 

Some minor comments: 

In Line 77: The authors write that IRF8 functions as repressor or activator dependent on IAD-associated factors. It would be nice to mention some examples of associated proteins for each case.

In line 94: It is mentioned that IRF8 can form complexes with ETS transcription factor. It should be noted, that the most important ETS factor in this context is PU.1. 

In line 53: IRF8 is presented as a tumor suppressor . Recent work suggests that IRF8 may function as oncogene in AML, where IRF8 is highly expressed. This should be mentioned, to make clear that IRF8 not exclusively acts as a tumor suppressor. 

In line 308: "IRF8 was discovered as a novel IGH partner." It should be made clear that a fusion partner is meant. 

In line 378: "IRF8 expression was associated with a better prognosis". Please make clear that high expression is meant.

Line 385-389: This statement goes too far, because it implies that IRF8 is downregulated in all human cancer cells. This should be revised. 

The cited paper only refers to lung cancer. IRF8 is highly expressed in other cancer types, such as DLBLC and AML. See also: http://gepia.cancer-pku.cn/detail.php?gene=IRF8

Line 460: "IRF8 ...  binds to the ISRE elements at the Spp1 promoter to repress OPN expression". The cited reference 27 does neither show a binding of IRF8 to the SPP1 promoter, nor that IRF8 functions as a repressor in this context. Presumbly reference 169 is meant. 

Figure 2: 

The used dataset "Cross-tissue immune cell analysis reveals tissue-specific features in humans" (reference 66) does NOT show the expression of IRF8 in organs but of immune cells within the organs. Thus, the authors should use a more appropriate dataset for making the claim that IRF8 is expressed in other tissues. (Such as from Proteinatlas or GTEx). B) These are no violin plots, but box plots. 

Figure 4B: These are no violin plots, but box plots. 

Author Response

Comment 1: However, the text often appears like a patchwork of not well connected statements and is not always easy to follow. Sometimes the review contradicts itself. For examples at the beginning it is written that IRF8 is not expressed in cancer cells, but later it is stated that IRF8 is often highly expressed in DLBLC. The review may therefore profit from some rewriting.

Revision 1: We than the reviewer for this advice and revised the entire manuscript, including txt and figures, as advised. The specific revisions are detailed below.

Comment 2: In Line 77: The authors write that IRF8 functions as repressor or activator dependent on IAD-associated factors. It would be nice to mention some examples of associated proteins for each case.

Revision 2: We added both activated and repressed genes advised.

Comment 3: In line 94: It is mentioned that IRF8 can form complexes with ETS transcription factor. It should be noted, that the most important ETS factor in this context is PU.1.

Revision 3: Clarified PU.1 as the most important ETS factor here.

Comment 4: In line 53: IRF8 is presented as a tumor suppressor. Recent work suggests that IRF8 may function as oncogene in AML, where IRF8 is highly expressed. This should be mentioned, to make clear that IRF8 not exclusively acts as a tumor suppressor.

Revision 4: Revised as advised. IRF8 has been reported as an AML suppressor and a AML oncogene.

Comment 5: In line 308: "IRF8 was discovered as a novel IGH partner." It should be made clear that a fusion partner is meant.

Revision 5: Revise as advised

Comment 6: In line 378: "IRF8 expression was associated with a better prognosis". Please make clear that high expression is meant.

Revision 6: Revise as advised

Comment 7: Line 385-389: This statement goes too far, because it implies that IRF8 is downregulated in all human cancer cells. This should be revised.

Revision 7:  Revised as advised. We now clarified that this is only in certain types of human cancers, and the scRNA data is in lung cancer.

Comment 8: The cited paper only refers to lung cancer. IRF8 is highly expressed in other cancer types, such as DLBLC and AML. See also: http://gepia.cancer-pku.cn/detail.php?gene=IRF8

Revision 8: As mentioned above, we have clarified that low expression of IRF8 in lung cancer.

Comment 9: Line 460: "IRF8 ...  binds to the ISRE elements at the Spp1 promoter to repress OPN expression". The cited reference 27 does neither show a binding of IRF8 to the SPP1 promoter, nor that IRF8 functions as a repressor in this context. Presumbly reference 169 is meant.

Revision 9: Corrected. Thanks.

Comment 10: Figure 2: The used dataset "Cross-tissue immune cell analysis reveals tissue-specific features in humans" (reference 66) does NOT show the expression of IRF8 in organs but of immune cells within the organs. Thus, the authors should use a more appropriate dataset for making the claim that IRF8 is expressed in other tissues. (Such as from Proteinatlas or GTEx). B) These are no violin plots, but box plots.

Revision 10: We replaced this figure with a new figure made with dataset from the Human Protein Atlas as advised.

Comment 11: Figure 4B: These are no violin plots, but box plots.

Revision 11: Corrected. Thanks.

Reviewer 2 Report

The review article entitled ‘IRF8: mechanism of action and health implications’ is very interesting and it describes role of IRF8 as a tumor suppressor and its role in immunologic responses orchestrated via various immune cells. However, The review article can be improved. Below are some comments.

1.      Can the authors describe if other abberations such as mutations or truncations are common in the loss of IRF8 ?

2.      The authors need to summarize the latest developments in the field.

3.      As loss of IRF8 has been described, what could be the possible reason.

4.      The possible degraders of IRF8 that can be targeted can be described in the article. Any post tranlational mechanisms of importance can be described. 

5.      More elaborate discussion is required for the potential role of IRF8 in Immunotherapy.

6.      The authors could elaborate on the future therapeutic interventions that could be targeted to rescue the expression of  IRF8  ?

7.      An overview picture of the various roles IRF8 plays in various biological process would be more illustrative, to summarize the roles of IRF8.

Author Response

Reviewer 2:

Comment 1: Can the authors describe if other abberations such as mutations or truncations are common in the loss of IRF8?

Revision 1: We have added IRF8 promoter polymorphisms and IRF8 somatic mutations.

Common 2: The authors need to summarize the latest developments in the field.

Revision 2:  We have updated reference and added the updated function of IRF8.

Common 3: As loss of IRF8 has been described, what could be the possible reason.

Revision 3: We have added/updated DNA methylation and miRNA as IRF8 loss mechanism.

Common 4: The possible degraders of IRF8 that can be targeted can be described in the article. Any post tranlational mechanisms of importance can be described.

Revision 4: We have added reviews of literature of IRF8 phosphorylation, ubiquitination and miRNA regulation.

Common 5: More elaborate discussion is required for the potential role of IRF8 in Immunotherapy.

Revision 5: We added the newest advance of IRF8 and immunotherapy in the conclusion part.

Common 6: The authors could elaborate on the future therapeutic interventions that could be targeted to rescue the expression of IRF8?

Revision 6: As mentioned above, we added a paragraph of IRF8 and immunotherapy.

Common 7: An overview picture of the various roles IRF8 plays in various biological process would be more illustrative, to summarize the roles of IRF8.

Revision 7: This is summarized in the graphic abstract.